# Assessing a behavioral nudge on healthcare leaders' intentions to implement evidence-based practices

Maia Crawford[1]*, A. James O'Malley[1,2], Ellen Meara[1,3], Taressa K. Fraze[4], Amber E. Barnato[1,5]

1 The Dartmouth Institute for Health Policy & Clinical Practice, Geisel School of Medicine at Dartmouth, Lebanon, NH, United States of America, 2 The Department of Biomedical Data Sciences, Geisel School of Medicine at Dartmouth, Lebanon, NH, United States of America, 3 The Department of Health Policy and Management, Harvard T.H. Chan School of Public Health, Boston, MA, United States of America, 4 Department of Family and Community Medicine, University of California, San Francisco, San Francisco, CA, United States of America, 5 Section of Palliative Care, Department of Medicine, Geisel School of Medicine at Dartmouth, Lebanon, NH, United States of America

* maia.l.crawford@dartmouth.edu

**Data Availability Statement:** Data cannot be shared publicly because confidentiality was guaranteed to NSHOS respondents. A subset of NSHOS practice- and hospital-level de-identified

## Abstract

### Importance

Leaders of healthcare organizations play a key role in developing, prioritizing, and implementing plans to adopt new evidence-based practices. This study examined whether a letter with peer comparison data and social norms messaging impacted healthcare leaders' decision to access a website with resources to support evidence-based practice adoption.

### Methods

Pragmatic, parallel-group, randomized controlled trial completed from December 2019 – June 2020. We randomized 2,387 healthcare leaders from health systems, hospitals, and physician practices in the United States, who had previously responded to our national survey of healthcare organizations, in a 1:1 allocation ratio to receive one of two cover letter versions via postal mail (all) and email (for the 60.6% with an email address), accompanying a report with their survey results. The "nudge" letter included messaging that highlighted how an organization's results compared to peers using text, color, and icons. Both nudge and control letters included links to a resource website. We interviewed 14 participants to understand how the letter and report impacted behaviors.

### Results

Twenty-two of 1,194 leaders (1.8%) sent the nudge letter accessed online resources, compared to 17 of 1193 (1.4%) sent the control letter ($p = 0.424$). Nine of the 14 interviewed leaders stated that viewing the letter (regardless of version) and accompanying report influenced their decision to take a subsequent action other than accessing the website. Seven leaders forwarded the report or discussed the results with colleagues; two leaders stated that receiving the letter and report resulted in a concrete practice change.

data, as well as de-identified datasets used to produce the findings in this study, are publicly accessible here: https://doi.org/10.3886/E165241V2.

**Funding:** This work was supported in part by the Agency for Healthcare Research and Quality's (AHRQ's) Comparative Health System Performance Initiative under a grant [1U19HS024075] to EM. It was also supported by the Levy Cluster in Healthcare Delivery at Dartmouth College in the form of a philanthropic endowment to AEB.

**Competing interests:** The authors have read the journal's policy and have the following competing interests: The statements, findings, conclusions, views, and opinions contained and expressed in this article are based in part on data obtained under license from IQVIA information services (OneKey subscription information services 2010-2017, IQVIA incorporated all rights reserved). There are no patents, products in development or marketed products associated with this research to declare. This does not alter our adherence to PLOS ONE policies on sharing data and materials.

## Conclusions

Receiving cover letters with a behavioral nudge did not increase the likelihood that organizational leaders accessed a resource website. Qualitative results suggested that the survey report's peer comparison data may have been a motivator for prioritizing and delegating implementation activities, but leaders themselves did not access our online resources.

## Introduction

Health professionals in the U.S. strive to deliver high-quality care that reflects the most up-to-date science available, yet they face numerous challenges integrating evidence-based practice into care delivery: insufficient knowledge, competing priorities, lack of support or authority, and resource constraints [1–3]. Health systems have increasingly sought to overcome these challenges by using behavioral science-inspired nudges to guide clinicians toward providing higher quality care. Nudging strategies, which aim to promote certain behaviors without restricting choice, include framing information in a motivating way and changing default choices from opt-in to opt-out [4–6]. Nudges have been successfully used in healthcare settings to increase generic medication prescribing rates, decrease imaging tests ordered at the end of life, and reduce the number of pills ordered per opioid prescription [7–9]. Recent systematic reviews of clinician-directed nudges affirm that nudges can improve clinical decision-making and effectively promote adherence to evidence-based clinical and administrative guidelines [10–15].

Studies aimed at overcoming barriers to evidence-based practice adoption, including via nudges, tend to focus on changing clinicians' perceptions and behaviors; yet it is administrative healthcare leaders–such as chief medical officers, chief quality officers, and practice mangers–who often initiate and oversee new practice implementation. These leaders use their position of authority to draw attention to improvement opportunities and guide the implementation process, playing an outsized role in facilitating change and influencing others' behaviors [16, 17]. Evidence suggests that influential leaders can have a sizeable impact on improving healthcare workers' compliance with evidence-based practice [18–21]. In fact, of the six organizational features recognized as influencing evidence-based practice adoption in healthcare settings, leadership was the only feature able to influence each of the others, indicating its central role in boosting or hindering evidence-based practice implementation [22].

Few studies have explored the mechanisms by which healthcare leaders themselves are influenced to act, and none to our knowledge have sought to directly influence healthcare leaders' choices through a behavioral nudge. We sought to devise an experiment to test the efficacy of nudging individuals who serve as "decision architects" for their respective organizations, delivering a more "upstream" intervention to change clinical practice behavior not at the clinician level, but at the decisionmaker level [23]. This work assumes that healthcare leaders maintain an "intention-behavior gap" related to evidence-based practice adoption: a discrepancy between what they plan or aspire to do and what they, by way of their organization, do in practice [24–26]. A number of existing frameworks help explain why decisionmakers' actions may not always align with their intentions, including the Theory of Planned Behavior, which describes how a person's actions are related to their beliefs about the behavior's consequences, others' expectations, and inhibiting or enabling factors [27]. Subsequent behavioral frameworks consider additional factors that influence behavior, like infrastructures, the political environment, and economics [28].

Nudges can help those interested in implementing evidence-based practices overcome organizational constraints. They do this by altering aspects of one's decision-making environment (known as "choice architecture") [6] to make implementation feel easier or more salient; this can be done by changing the placement or attributes of objects or modifying messaging [5, 29, 30]. Nudging is therefore one strategy among many to promote or strengthen implementation science efforts, which encourage the uptake of evidence-based practices to improve the quality and efficacy of health services [31, 32]. To test the efficacy of nudging healthcare leaders to work toward promoting organizational adoption of recommended care delivery practices, we randomized leaders to receive a standard cover letter versus a "nudge" cover letter, both accompanying a peer comparison report. The nudge cover letter was inspired by prior studies that used social norms messaging to establish standards or expectations around a specific behavior, such as overprescribing certain medications [33–37].

## Methods

### Study design

We conducted a pragmatic, parallel-group, randomized controlled trial comparing the effects of two cover letter versions on organizational leaders' behavior between December 2019 and February 2020. The cover letters accompanied a personalized survey report that was mailed via the U.S. Postal Service to all respondents of the National Survey of Healthcare Organizations and Systems (NSHOS). The NSHOS was a suite of nationally representative surveys to characterize the structure, ownership, leadership, and care delivery capabilities of healthcare systems, physician practices, and hospitals, funded by the U.S. Agency for Healthcare Research and Quality (AHRQ). The NSHOS, fielded by Dartmouth and collaborators in 2017–2018, included a protocol to share study results. This was done through the development of peer comparison reports, which included information about how each NSHOS participant responded to a subset of survey questions, as well as how peer organizations responded (S1 File). In addition to mailing the cover letter and report, we emailed all NSHOS respondents for whom we had an email address, and provided them with a hyperlink to access their personalized cover letter and report using a passcode contained in the same e-mail.

The study was approved by the Committee for the Protection of Human Subjects at Dartmouth (Study 28763) and followed the CONSORT Reporting Guidelines [38]. NSHOS respondents provided written informed consent when completing the survey; qualitative interview participants provided verbal consent to the interviews. The study was also submitted to clinicaltrials.gov (Identifier: NCT04176146).

### Study population

Study participants were healthcare organization leaders from health systems, hospitals and physician practices who responded to the NSHOS in 2017 and 2018 [39]. There were 3,402 unique responses from the 7,392 organizations surveyed. This study relied exclusively on the NSHOS study sample and associated contact information. The NSHOS sample frame was developed using data from the 2015 IQVIA OneKey database, which included organizations' physical and postal addresses, along with supplemental data from the American Hospital Association and AHRQ. Once organizations were sampled, research assistants worked to verify the data provided and gather names, email addresses, and phone numbers of organizational leadership.

Chief Executive Officers, Chief Medical Officers, and Chief Clinical Officers were targeted to fill out the NSHOS from health systems and hospitals, with health system Chief

Administrative Officers and Chief Clinical Officers and hospital Chiefs of Medicine and Medical Staff serving as secondary targets. Practice managers, administrators, or lead physicians were targeted to fill out the survey at physician practices; secondary targets were currently practicing primary care physicians. Eligibility for the trial was determined by whether an NSHOS respondent reported that their organization had not adopted at least one of up to seven pre-determined care delivery practices. Of the 3,402 assessed organizations, 2,387 (70.2%) reported not adopting at least one of the chosen care delivery practices and were therefore eligible for the study (S2 File).

## Randomization

The study statistician used a computerized random number generator to randomize the 2,387 leaders in a 1:1 allocation ratio to receive their peer comparison reports with either a "nudge" or control cover letter (Fig 1).

## Intervention

To develop the intervention (or nudge) cover letter, the study team pilot tested draft materials and dissemination strategies with two high-level health system executives, one health services researcher with expertise in behavioral nudges, and a Dartmouth research team that develops and tests patient and provider decision support tools, iteratively modifying materials after each round of input. We also received feedback on materials to include on the resource webpages from 10 clinical subject matter experts.

The letters were addressed to specific individuals and signed by two Dartmouth researchers. Following a short introduction, the nudge letter stated: "[organization name] has not implemented [x] of seven common care delivery practices that the majority of your peers have already implemented." This statement used descriptive norms to directly compare the leader's organization to its peers, a strategy meant to demonstrate that evidence-based care practice adoption was the norm among similar organizations. It then included a table listing the seven evidence-based practices and indicating whether the leader's organization adopted each practice using a green thumbs up for adoption and a red caution triangle for non-adoption. The use of color coding and positive and negative icons employed injunctive norms–the inference of others' validation or support–to convey approval for adoption and disapproval for non-adoption (S3 File) [40, 41]. The table also included peer comparison data for the seven featured care delivery practices, using horizontal bars to display the adoption percentage for each care practice among peer organizations. To the right of the peer comparison data was a hyperlink to access "tools to enhance care practice adoption."

The control letter included the same introductory paragraph, but without the statement noting the organization's number of non-adopted practices. Like the nudge letter, the control letter had a table listing the seven featured care practices, but this table did not include data about whether the organization adopted each practice (and therefore, no colors or icons), nor did it contain any peer comparison data. It only included the same hyperlink to online tools (S4 File).

Both the intervention and the control letter versions included a hyperlink to a website that contained pages with curated, expert-validated resources for download, meant to help healthcare organizations move toward adopting the care delivery practices. Resources on the website included checklists, physician pocket cards, surveys, evidence reviews, and clinical practices guidelines (S5 File). The intervention and control letters had different hyperlinks for ease of tracking website views, though the websites were identical.

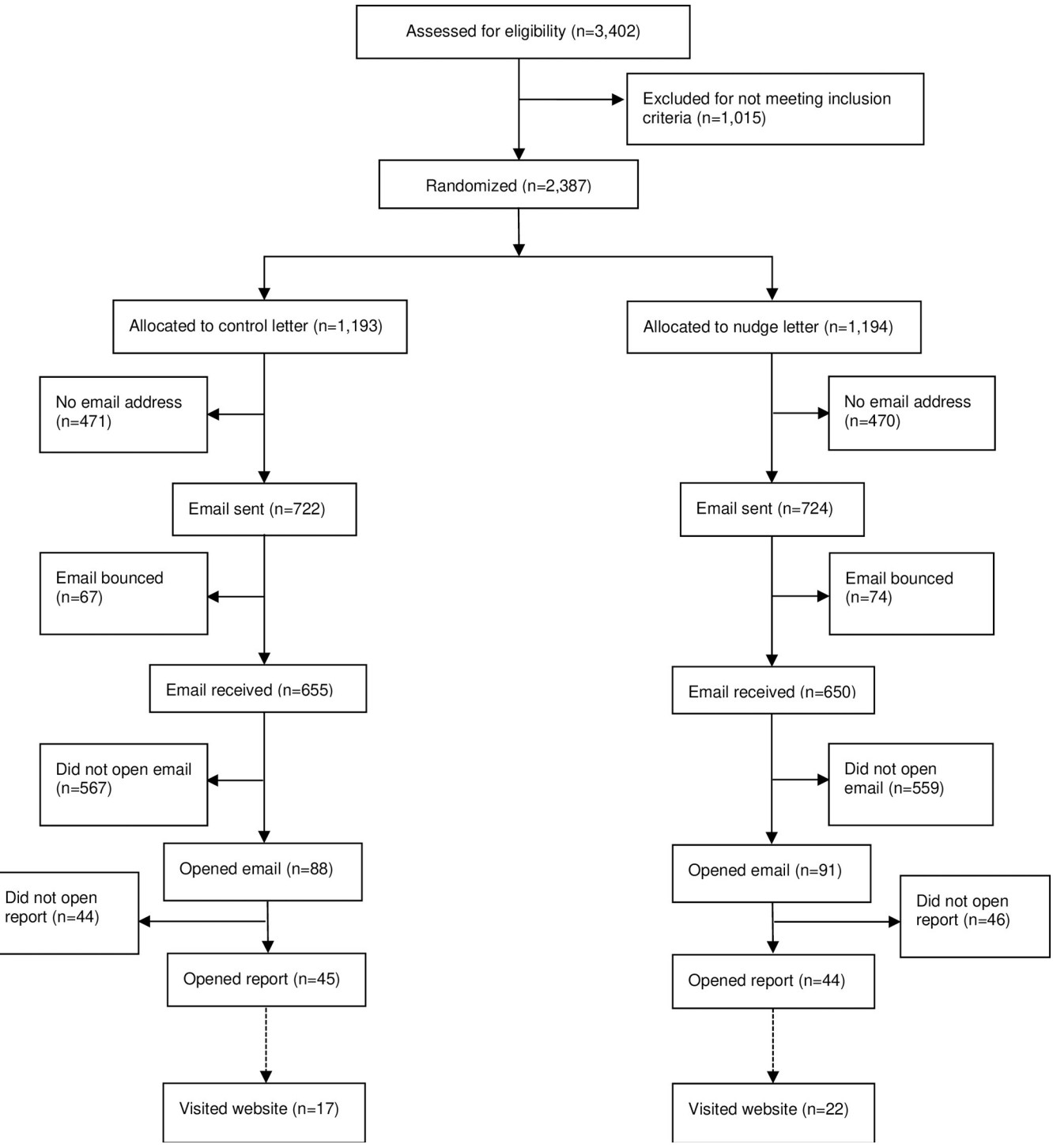

**Fig 1. Flow diagram of emailed letter dissemination.** The dashed line between "opened report" and "visited website" indicates that we cannot be certain that all website viewers were study participants who received the email version of the letter.

## Intervention delivery

Hard-copy cover letters and survey reports were mailed via the U.S. Postal Service to all 2,387 study participants between December 6 and December 10, 2019. Cover letters and survey

reports were additionally emailed to all organizations with an email address on file on December 18, 2019 (n = 1,446, 60.6% of study participants). E-mail recipients had to click on a hyperlink and enter a unique, five-character password contained in the same e-mail to access a downloadable pdf of their personalized letter and report. We then sent two additional emails to individuals who had not opened the previous email(s) on January 8 and January 20, 2020 (S6 and S7 Files). The subject line for the first email was "See how you compare: Your AHRQ-funded survey results." Subject lines for the two subsequent reminder emails were: "Thank you for completing the AHRQ-funded survey; See your results" and "AHRQ-funded survey results just released; see how you compare." We did not pilot test these e-mail subject lines.

## Outcome measures

The primary outcome measure was the percentage of all study participants who accessed the technical assistance website (intention-to-treat analysis), as determined by Google Analytics. We also included a per-protocol analysis to assess website access among participants who received the letter (i.e., the mailed letter was deliverable and/or the email was deliverable). Secondary outcomes were: unique resource downloads; requests to be connected to a peer organization for further information-sharing (an option on the resource website); and the perceived effects of the letter on administrators' subsequent actions and intentions, as measured by qualitative interviews.

## Statistical analysis

We used Google Analytics to measure the number of visits to the intervention and control resource websites from December 9, 2019 –February 9, 2020. We used two-sample $Z$-tests for differences in population proportions to evaluate whether observed differences in website viewing behavior between groups were statistically significant. Assuming an estimated rate of accessing online resources in the control group of 5%, the study was designed to have 80% statistical power to detect a 3-percentage point increase in the probability of accessing online resources in the intervention group.

## Qualitative interviews

Between March and June 2020, we emailed each of the 89 individuals who electronically downloaded their survey reports (44 in the nudge arm and 45 in the control arm) to request participation in a qualitative interview. Our outreach coincided with the onset of the COVID-19 pandemic and many healthcare leaders did not respond to our email or declined our request due to pandemic-related responsibilities. Fourteen organizational leaders agreed to participate: eight who electronically viewed the nudge letter and six who electronically viewed the control letter. The 14 interviewees were associated with five physician practices, five hospitals, and four healthcare systems.

The goal of these interviews was to better understand if receipt of the study materials spurred additional actions or intentions beyond accessing the resource website. The interview guide included questions related to: leaders' reaction to the letter and report, past and future actions that could be attributed to receiving the letter and report, and motivators for adopting new care delivery practices (S8 File). Interviews were audio-recorded and then professionally transcribed [42]. One investigator read and coded all transcripts to generate a list of themes; a different investigator independently reviewed themes and exemplar quotes [43, 44].

## Results

### Organization characteristics

Organizations eligible for the nudge intervention were diverse in their structure and operations. Of the 2,387 organizations eligible for this study, 494 were hospitals, 1,627 were physician practices, and 266 were healthcare systems. Overall, 803 had not implemented one of the featured evidence-based care practices; 646 had not implemented two of the practices; 428 had not implemented three of the practices; and 510 had not implemented four or more of the practices (Table 1).

Among the eligible organizations, 1,194 were sent the nudge letter and 1,193 were sent the control letter. A total of 5.2% of the mailed letters were undeliverable. Among the 1,446 eligible organizations with an e-mail address on file (who received an e-mailed link to an electronic version of the cover letter and report in addition to the mailed versions), 724 were sent the nudge letter and 722 were sent the control letter. Among the 1,305 with a working email address, 14.0% of nudge arm participants opened their emails and 6.9% downloaded their report; 13.4% of control arm participants opened their emails and 6.7% downloaded their report.

### Primary outcome

We summarize outcomes in Table 2. Among all 2,387 organizational leaders included in the study sample and the intention-to-treat (ITT) analysis, 22 (1.8%) in the nudge arm and 17 (1.4%) in the control arm accessed the resource website, a difference of 0.4 percentage points (-0.6, 1.4).

To aid the interpretation of this effect size, we transformed it to the hypothetical situation in which only those leaders with a working email address accessed the resource website

**Table 1. Characteristics of study participants.**

| | Overall (N = 3,402) | Control (N = 1,193) | Treatment (N = 1,194) | Ineligible (N = 1,015) |
|---|---|---|---|---|
| **Organization Type, No. (%)** | | | | |
| Healthcare system | 432 (13) | 132 (11) | 134 (11) | 166 (16) |
| Complex integrated system | 90 (3) | 30 (3) | 30 (3) | 30 (3) |
| Simple integrated system | 177 (5) | 50 (4) | 51 (4) | 76 (7) |
| Medical group | 165 (5) | 52 (4) | 53 (4) | 60 (6) |
| Hospital | 743 (22) | 248 (21) | 246 (21) | 249 (25) |
| General acute care hospital | 485 (14) | 161 (14) | 160 (14) | 164 (16) |
| Academic medical center | 64 (2) | 15 (1) | 13 (1) | 36 (4) |
| Critical access hospital | 194 (6) | 72 (6) | 73 (6) | 49 (5) |
| Physician practice | 2,227 (65) | 813 (68) | 814 (68) | 600 (59) |
| Small (less than 5 physicians) | 1,140 (33) | 427 (36) | 429 (36) | 284 (28) |
| Medium (6–10 physicians) | 534 (16) | 192 (16) | 192 (16) | 150 (15) |
| Large (11+ physicians) | 553 (16) | 194 (16) | 193 (16) | 166 (16) |
| **With email address on file, No. (%)** | 2,071 (61) | 722 (61) | 724 (61) | 625 (62) |
| **Unimplemented evidence-based care practices, No. (%)** | | | | |
| 0 | 1,015 (30) | 0 (0) | 0 (0) | 1,015 (100) |
| 1 | 803 (24) | 401 (34) | 402 (34) | 0 (0) |
| 2 | 646 (19) | 323 (27) | 323 (27) | 0 (0) |
| 3 | 428 (12) | 214 (18) | 214 (18) | 0 (0) |
| 4+ | 510 (15) | 255 (21) | 255 (21) | 0 (0) |

**Table 2. Study outcomes, by group.**

| | Control Arm | Treatment Arm | Difference (95% CI), pp |
|---|---|---|---|
| **No. who accessed website / No. eligible (%)** | | | |
| **All recipients** | 17/1193 (1.4) | 22/1194 (1.8) | 0.4 pp (-0.6, 1.4) |
| **All recipients with letter or email delivered** | 17/1157 (1.5) | 22/1168 (1.9) | 0.4 pp (-0.6, 1.5) |
| **All recipients with an email address[*]** | 17/722 (2.4) | 22/724 (3.0) | 0.6 pp |
| **All recipients with an email address that did not bounce back[*]** | 17/655 (2.6) | 22/650 (3.4) | 0.8 pp |
| **All recipients who opened emailed report[*]** | 17/45 (37.8) | 22/44 (50) | 12.2 pp |
| **Unique website resource views/downloads, No.** | 33 | 16 | |
| **Requested Connection to Peer Organizations, No.** | 1 | 1 | |

[*] The proportions reported in subgroups with an asterisk assume that all website visits originated from the subgroup. In particular, these proportions assume that the only recipients were those who were able to click on the emailed hyperlink rather than typing in the hyperlink from the paper letter.

CI = Confidence interval

pp = percentage point

(n = 1,305), a condition we hypothesize represents real-world circumstances, based on research about online behaviors and qualitative data suggesting leaders did not manually type in the website address from the paper letter [45]. Under this hypothetical scenario, we would calculate that 3.4% of leaders in the nudge arm accessed the website vs. 2.6% in the control arm (difference of 0.8 percentage points). Therefore, the maximum effect of receiving the nudge via email is estimated to be 0.8 percentage points, double the ITT estimate. Because this is a hypothetical calculation, no confidence interval is provided; it would be approximately equal to that computed above.

## Secondary outcomes

As summarized in Exhibit 2, a total of 49 resources were viewed or downloaded; 16 were viewed on the nudge website and 33 on the control website (21 of the 33 control site downloads resulted from one visitor). Two website visitors requested a connection to a peer organization (an option available on the resource webpage): one nudge website visitor and one control website visitor.

## Qualitative insights on receiving the cover letter and report

Findings from the semi-structured interviews with 14 organizational leaders who electronically viewed the letter and report are summarized in Table 3. While just two of 14 interviewees remembered accessing the resource webpage, nine of the interviewed leaders remembered taking some other subsequent action after receiving the cover letter and report. This was true regardless of which letter version they received. In most cases, the leaders either forwarded the report to colleagues, such as to the organization's Director of Quality or Chief Nursing Officer, or initiated conversations with colleagues to discuss the results or opportunities to address perceived deficiencies (n = 7).

Within two organizations (one nudge, one control), interviewees indicated that receipt of the report directly contributed to concrete programmatic or structural changes within a three-to-four-month timeframe: one organization adopted a new team-based rounding process to review patients with complex clinical needs; a second organization altered its electronic medical record platform so clinicians could more easily access an evidence-based depression screening tool. Five of eight nudge letter recipients and four of six control letter recipients also noted that the report may lead to future organizational changes, namely by serving as a

**Table 3. Qualitative interview findings.**

| Theme | Specific Findings | Exemplar Quotes |
|---|---|---|
| **First reactions to cover letter and report** | Drawn first to the cover letter "no" responses (nudge letter recipients) (nudge = 8/8)<br>Found the nudge messaging and peer comparison data impactful or persuasive (nudge letter recipients) (nudge = 6/8)<br>Thought responses not representative of current circumstances<br>(nudge = 4/8; control = 4/6) | "I don't generally consider myself to be a negative, glass is half empty kind of person, but I immediately went to the one that we did not [implement]. I. . . was surprised and disappointed and then disbelieving." (*Hospital leader, nudge letter*)<br>"I realized how quickly things change. [. . .] Because some of the responses may have been accurate two years ago, but aren't accurate today just because things are so dynamic." (*Health system leader, control letter*)<br>"Seeing a "no" response when peers had a "yes" was a motivator" (*Physician practice leader, nudge letter*)<br>"My eyes immediately went to the thumbs up and the red triangle." (*Hospital leader, nudge letter*) |
| **Examples of actions taken after viewing letter and report*** | Forwarded report to colleagues and/or discussed results (nudge = 4/8; control = 3/6)<br>Initiated new rounding approach to develop care plans for complex patients (nudge = 1/8; control = 0/6)<br>Signed up for a webinar (nudge = 1/8; control = 0/6)<br>Changed electronic health record format to enable faster access to depression screening questionnaires (nudge = 0/8; control = 1/6) | "And so I think this did help to give us that little [nudge] in the right direction to move forward [to initiate a new complex care management program]." (*Physician practice leader, nudge letter*)<br>"I also passed it to our quality management department. [. . .] We have a lean six sigma performance improvement process that picks out practices like this every year, a handful that we focus on. . . and I just wanted them to be aware of [the report]. (*Hospital leader, nudge letter*)<br>"[The letter and report] help[ed] heighten the fact that we needed to do something better. And it kind of restarted something we started way back when, because this trying to get the PHQ-2 moved up in Epic was something that was started five years ago, but the project got lost on the Epic build team. So it was kind of like, "Hey guys, what happened to this?" And we were successful then to get it moved forward." (*Hospital leader, control letter*) |
| **Examples of anticipated actions** | Use results to help set or change quality improvement priorities and reform efforts (nudge = 1/8; control = 4/6)<br>Gather more information about status quo and explore resources/tools to work toward implementation (nudge = 4/8; control = 0/6) | "I'm actually really intrigued to look into more detail about the method for identifying complex, high need patients. That really sparked my attention to investigate and look at that a little bit more. So I definitely see us looking into that." (*Hospital leader, nudge letter*)<br>"I would really be able to use this as a conversation starter for quality initiatives. [. . .] Lots of people are bringing things up that they want. If they have a particular interest in X and would like to see us do more of that, I can say, "Actually we're doing it," or I would say, "That didn't come up on the list. So, maybe it's a lower priority." (*Physician practice leader, nudge letter*) |
| **Motivators for adopting evidence-based care practices** | High quality patient care (nudge = 6/8; control = 5/6)<br>Comparison to peers (nudge = 6/8; control = 5/6)<br>Cost considerations (nudge = 4/8; control = 3/6)<br>Internal leadership (nudge = 1/8; control = 2/6)<br>Regulatory/contract requirements (nudge = 1/8; control = 1/6) | "Two equal motivations: Are we doing something to improve the quality care that we're delivering to the patients that we serve? And are we going to do something that's going to help them reduce the overall cost of care with the system?" (*Physician practice leader, control letter*)<br>"[I] want to make sure that we are continuing to stay ahead of the curve or at least in line with what others are offering their patients." (*Physician practice leader, nudge letter*) |

* Not mutually exclusive; one interviewed leader who received the control letter indicated both forwarding the report to colleagues and changing their organization's electronic health record format.

"conversation starter" for setting quality improvement priorities and focusing attention on specific practices to adopt. Regardless of the study arm, most interviewees (11 of 14) said data on peer organizations' performance was somewhat or very motivating.

## Discussion

This pragmatic randomized trial evaluated whether letters with behavioral science-informed messaging and data increased the likelihood that an organizational leader would access a resource website, which we viewed as a proxy for their intention to work toward adopting new evidence-based practices. There were no statistically significant effects of the intervention on

website access. Regardless of cover letter format, website access was low, with 1.6% of all organizational leaders targeted clicking through to website resources (3.0% of leaders with a working email address).

There are many potential reasons for the low response to the information presented in the nudge letter, including that the primary outcome variable (resource website views) did not accurately capture a leader's intention to implement a new care practice. It is possible that nudge letter recipients were more motivated than control letter recipients to work toward adopting new care practices, despite not accessing online resources. Our interviewees suggested that not visiting the website was not associated with their intention (or lack thereof) to adopt new care practices; instead, leaders did not click because they did notice the link, did not have the time, or did not think doing so aligned with their professional responsibilities. The pragmatic decision made by the study team to use website views as a proxy for the underlying construct of interest, "intention to act," therefore introduced an inherent incongruity between our study targets' professional role and their standard work behavior. We therefore conclude that website views were an imperfect representation of a leader's intention to promote new care practice adoption.

Additionally, certain nudge letter features, including its source (academic researchers), and relatively understated messaging, may not have been compelling or persuasive enough to engender action. Past letter-based nudge studies with positive results sent multiple letters from high-profile or authoritative figures (e.g., England's Chief Medical Officer) and warned of the negative consequences of inaction (e.g., a federal audit) [33, 46]. Based on our findings, we do not believe that those who received the nudge letters were more "primed" to act than those receiving the control letters. When the quantitative and qualitative results are interpreted together, we find that the nudge letter recipients did not take more or different actions than control letter recipients: a similar number within both groups visited the website, forwarded the reports, and took other subsequent actions. While our qualitative data likely reflect the perceptions of particularly motivated and engaged leaders, the fact that nine of 14 interviewed leaders shared the materials with colleagues or initiated other subsequent activities suggests the peer comparison reports themselves provided a baseline incentive to act, while the social norms messaging in the nudge letter may not have provided an additional marginal incentive.

This interpretation aligns with interviewees' belief that receiving peer comparison data related to evidence-based practice adoption was somewhat or very motivating. The peer comparison data may have been particularly compelling because they pertained to the quality of patient care; literature suggests that healthcare workers feel a strong moral obligation (i.e., intrinsic motivation) to provide the highest standard of care possible [47]. While peer comparison data as an information framing strategy is not considered a particularly strong nudge, such data can produce desired effects if used thoughtfully and strategically [12, 48]. The fact that our study subjects–data-driven, outcomes-oriented, business-minded individuals–appeared more influenced by the peer comparison data demonstrates the importance of accounting for audience characteristics and situational context when applying nudge theory and designing nudge strategies. In retrospect, use of peer comparison data seems highly appropriate for leaders who need to make reflective professional decisions; social norms messaging and iconography–which produce automatic, almost subconscious reactions–may be more effective nudge strategies for individuals who need to make quicker, simpler decisions.

The intervention's low impact could also relate to weaknesses in how the letter was designed and delivered. First, many leaders never received or opened their letter and report. While we do not know the open rate for the paper letters–an inherent shortcoming of that delivery modality–the 13.7% email open rate is at the low end of the 9.3%– 46.0% range for studies in which physicians were emailed to take an online survey or view web-based materials

(we are unaware of comparable data for healthcare executives or similar business leaders) [49–53]. Our study's low open rate suggests that the email's subject line, which was not personalized, did not capture the attention of busy, high-ranking healthcare leaders who receive hundreds of emails per day; the emails may also have been dismissed as spam or marketing [54, 55]. Among those who opened the email, about half accessed their reports. Removing the additional passcode entry step would likely further increase report download rates. Finally, we sent the letters in December 2019 and January 2020, a time when leaders may have been busy with end-of-year activities, budget planning, vacations, and early preparations for the COVID pandemic.

## Strengths and limitations

This study is the first that we know of to use nudging strategies to influence administrative healthcare leaders' intentions and actions to change organizational practice. We had a large study sample of over 2,300 administrative leaders from a range of healthcare settings and designed a study protocol using best practice randomization and allocation strategies. Our cover letters were developed following an extensive review of the literature on social norms messaging and written behavioral nudges [33, 35, 56–67]. We also shared the letters and resource website with a range of experts and modified our design and content based on their feedback.

This study also has several limitations. First, we were showcasing results from a self-reported survey that was filled out up to two-and-a-half years before the reports were disseminated. Results may therefore not have reflected current circumstances. Second, we were unable to assess how many people viewed the paper version of the letter, and of those who did, how many (if any) accessed the website. Third, all survey reports included peer comparison data, which could have influenced control group participants' decision to click on the resource website or initiate other implementation actions. Fourth, Google Analytics does not link website visits to specific users, so we were unable to determine which individuals viewed the site [68, 69]. Fifth, the qualitative interviews were conducted with a small sample of leaders who viewed their reports and agreed to be interviewed, suggesting a high degree of interest and engagement in the study topic, making it hard to draw broad-based conclusions applicable to all healthcare leaders from the qualitative results. Finally, we decided to include all eligible participants in the study to reach as many individuals as possible, but this decision impeded our ability to track paper letter receipt and align this delivery modality with the outcome of interest (website views).

## Implications

The present study offers new insights about targeting administrative healthcare leaders to plan and initiate care practice changes, as well as how to best construct and deliver materials for this audience. As we were unable to test the impact of our cover letters among a large group (only 89 of over 2,300 study subjects electronically viewed their letters and had known access to the website link), future studies targeting healthcare leaders should invest more time and resources in optimizing communication materials' design and dissemination. Opportunities to enhance email-based studies include using a more attention-grabbing email sender–such as a known local contact or a national authority figure–and obtaining more up-to-date and high-quality contact information. Additionally, researchers should personalize subject lines and be purposeful about scheduling email deliveries at times associated with higher open rates [70]. The study team should also invest in more intensive pilot testing, employing an iterative, user-

centered design approach of eliciting feedback and revising materials through successive rounds of testing [71, 72].

The qualitative results indicate that a subset of report viewers took actions other than accessing the website, suggesting that administrative healthcare leaders may be an appropriate and receptive audience to interventions meant to improve evidence-based practice adoption. These results also suggest that some healthcare leaders were receptive to and motivated by peer comparison data; researchers should consider further testing the use of peer comparison data among this group. De-coupling peer comparison data from other nudging strategies would help determine its efficacy in isolation.

## Conclusion

This study tested whether healthcare leaders were more likely to access a resource website after viewing a cover letter with behavioral science-inspired messaging and data display. We did not find a statistically significant difference in website views between those who received the nudge and control letters. The low email open and report download rates for all study subjects imply shortcomings in the execution of our letter-based experiment that likely led to low website viewing rates.

The quantitative and qualitative results together suggest that actions taken following the viewing of the letters and reports were unlikely to be associated with the cover letter messaging, as the two study arms' subsequent actions did not discernably differ. The fact that control letter recipients took any actions suggests that they may have been motivated by the data in their personalized reports, which compared survey responses to those from similar healthcare organizations. Future research that targets administrative healthcare leaders to improve organization-wide practice adoption should make purposeful design and dissemination choices; invest in user-centered pilot testing; and align interventions with leaders' motivations, priorities, and workflows.

## Supporting information

**S1 File. Survey peer comparison report.**
(DOCX)

**S2 File. Supplementary methods.**
(DOCX)

**S3 File. Nudge hard copy letter.**
(DOCX)

**S4 File. Control hard copy letter.**
(DOCX)

**S5 File. Resource website resource page screenshot.**
(DOCX)

**S6 File. Nudge email.**
(DOCX)

**S7 File. Control email.**
(DOCX)

**S8 File. Qualitative interview guide.**
(DOCX)

## Acknowledgments

Contributors: The authors thank Kristy Bronner and Emily Luy Tan for their research support.

## Author Contributions

**Conceptualization:** Amber E. Barnato.

**Formal analysis:** A. James O'Malley.

**Funding acquisition:** Amber E. Barnato.

**Investigation:** Maia Crawford, Amber E. Barnato.

**Methodology:** A. James O'Malley, Amber E. Barnato.

**Project administration:** Maia Crawford, Amber E. Barnato.

**Resources:** Maia Crawford.

**Supervision:** Maia Crawford, Ellen Meara, Taressa K. Fraze, Amber E. Barnato.

**Validation:** A. James O'Malley, Ellen Meara, Taressa K. Fraze, Amber E. Barnato.

**Visualization:** Maia Crawford.

**Writing – original draft:** Maia Crawford.

**Writing – review & editing:** Maia Crawford, A. James O'Malley, Ellen Meara, Taressa K. Fraze, Amber E. Barnato.

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
