## [Decision Letter · Decision Letter 0]

15 Jan 2024

PONE-D-23-33529Assessing a behavioral nudge on healthcare leaders’ intentions to implement evidence-based practicesPLOS ONE

Dear Dr. Crawford,

Thank you for submitting your manuscript to PLOS ONE. After careful consideration, we feel that it has merit but does not fully meet PLOS ONE’s publication criteria as it currently stands. Therefore, we invite you to submit a revised version of the manuscript that addresses the points raised during the review process. The reviewers have endeavored to provide numerous constructive suggestions to improve the article. Please take them into account. In particular, following the reviewers’ comments, please provide more details about how the intervention was delivered, as well as a more and better developped discussion. Please submit your revised manuscript by Feb 29 2024 11:59PM. If you will need more time than this to complete your revisions, please reply to this message or contact the journal office at plosone@plos.org. Please include the following items when submitting your revised manuscript:A rebuttal letter that responds to each point raised by the academic editor and reviewer(s). You should upload this letter as a separate file labeled 'Response to Reviewers'.A marked-up copy of your manuscript that highlights changes made to the original version. You should upload this as a separate file labeled 'Revised Manuscript with Track Changes'.An unmarked version of your revised paper without tracked changes. You should upload this as a separate file labeled 'Manuscript'.

We look forward to receiving your revised manuscript.

Kind regards,

Alberto Molina Pérez, Ph.D.

Academic Editor

PLOS ONE

“This work was supported in part by the Agency for Healthcare Research and Quality's (AHRQ's) Comparative Health System Performance Initiative under Grant # 1U19HS024075, which studies how healthcare delivery systems promote evidence-based practices and patient-centered outcomes research in delivering care (MC, JO, EM, TF, AB) (ahrq.gov). It was also supported by the Levy Cluster in Healthcare Delivery at Dartmouth College (AB) (https://sites.dartmouth.edu/levyincubator/).”

“Contributors: The authors thank Kristy Bronner and Emily Luy Tan for their research support.

Funders: This work was supported in part by the Agency for Healthcare Research and Quality's (AHRQ's) Comparative Health System Performance Initiative under Grant # 1U19HS024075, which studies how healthcare delivery systems promote evidence-based practices and patient-centered outcomes research in delivering care. It was also supported by the Levy Cluster in Healthcare Delivery at Dartmouth College. The statements, findings, conclusions, views, and opinions contained and expressed in this article are based in part on data obtained under license from IQVIA information services (OneKey subscription information services 2010-2017, IQVIA incorporated all rights reserved). The statements, findings, conclusions, views, and opinions contained and expressed herein are not necessarily those of IQVIA Inc. or any of its affiliated or subsidiary entities. The views presented herein do not represent those of the Federal Government.”

“This work was supported in part by the Agency for Healthcare Research and Quality's (AHRQ's) Comparative Health System Performance Initiative under Grant # 1U19HS024075, which studies how healthcare delivery systems promote evidence-based practices and patient-centered outcomes research in delivering care (MC, JO, EM, TF, AB) (ahrq.gov). It was also supported by the Levy Cluster in Healthcare Delivery at Dartmouth College (AB) (https://sites.dartmouth.edu/levyincubator/).”

5. In the online submission form you indicate that your data is not available for proprietary reasons and have provided a contact point for accessing this data. Please note that your current contact point is a co-author on this manuscript. According to our Data Policy, the contact point must not be an author on the manuscript and must be an institutional contact, ideally not an individual. Please revise your data statement to a non-author institutional point of contact, such as a data access or ethics committee, and send this to us via return email. Please also include contact information for the third party organization, and please include the full citation of where the data can be found.

Reviewers' comments:

Reviewer's Responses to Questions

**Comments to the Author**

1. Is the manuscript technically sound, and do the data support the conclusions?

Reviewer #1: Yes

Reviewer #2: Partly

2. Has the statistical analysis been performed appropriately and rigorously? 

Reviewer #1: Yes

Reviewer #2: No

3. Have the authors made all data underlying the findings in their manuscript fully available?

Reviewer #1: No

Reviewer #2: Yes

4. Is the manuscript presented in an intelligible fashion and written in standard English?

Reviewer #1: Yes

Reviewer #2: Yes

5. Review Comments to the Author

Reviewer #1: This study examined whether a letter with social norms messaging and peer comparison data

impacted healthcare leaders’ decision to access a website with resources to support

evidence-based practice adoption. The team conducted a pragmatic, randomized controlled trial of 2,387 healthcare leaders from health systems, hospitals, and physician practices in the United States to receive one of two cover letter versions, each linking to a resource website and accompanying a report with survey results. The “nudge” letter included messaging that highlighted how an organization’s results compared to peers using text, color, and icons. The letters were sent in Dec 2019 and emails and followup emails were sent Dec 2019 and Jan 2020 to those with email addresses. The team interviewed 14 participants to understand how the letter and report impacted behaviors. 1.8% of the leaders sent the nudge letter accessed online resources, compared to 1.4% sent the control letter (p=0.424).

Overall, this study makes an excellent contribution to several areas of the literature in terms of applied behavioral science and nudge interventions in health care, health care quality improvement, and implementation science. The nudge literature has been criticized for publication bias. It’s critical that rigorous randomized trials with null findings like this are published to better understand what works and what doesn’t. The nudge interventions were well conceived and it’s likely that many would have thought they could be effective at increasing engagement with quality improvement interventions. The results will be informative to quality improvement leaders as they plan future interventions to increase adoption of evidence-based practices.

I have some recommendations for improving the paper and it’s impact:

1. More clarity needed in description of how the intervention was delivered.

a. I had to read through the methods and the supplemented several times to understand how the intervention was delivered. It should be stated more clearly that the letters were sent by postal mail to all individuals who were randomized and that an email was sent to ~60% of the randomized individuals who had an email address. S1 Figure 1 makes this much more clear than Figure 1. S1 Figure 1 should be included in the main manuscript instead of the current Figure 1.

b. Given that over 86% of those who were emailed the letter didn’t open the email, it’s critical to know what the subject line of the email was. This should be stated in the manuscript and included in the supplement.

c. It should also be more clearly stated that the email required a login step to get into the report

2. Calling out important exploratory outcomes in the Results. This is mentioned in the discussion, but not called out in the Results.

a. The fact that unique website resources downloads was no different, and likely even more common in the control group is something that should be called out because that’s the final behavior the researchers are trying to improve. This is less that 2% overall of intended recipients, indicating that the efforts were unsuccessful for 98% of the targeted recipients.

3. Discussion and key lessons need to be reframed around the major findings of the trial.

a. Last line of the discussion 266-267 – “The qualitative results, however, suggest

that providing administrative leaders with data comparing their organization’s performance to that of peers can spur follow-up action aimed at promoting evidence-based practice adoption” needs to be reframed. These qualitative results were only collected among 2% of the intended recipients who engaged with the intervention. Would reframe the qualitative findings to be among the small proportion of the intended recipients who engaged with the intervention. Furthermore, the qualitative methods are limited and don’t include a conceptual model and with very little description of how the responses were coded and analyzed. I would worry that the quotes were cherry picked to add some positive findings around the intervention. I’m not sure that very much can be concluded from the qualitative analysis since engagement was so low and engagement is likely reflective of highly motivated individuals. This is even stated as a limitation at the end that “ the qualitative interviews were conducted with a small sample, so it is hard to draw broad-based conclusions from them.”

b. The stated three lessons from the results do not align closely with the key findings of the trial, particularly the first two: “1) peer comparison data can be powerful…and 2) Outcome variables should align.” The key finding of the study is that a letter distributed through postal mail and email was that 86% of health care leaders didn’t open the email, and only 1.4% of the leaders visited the website with implementation resources. The other key finding was that even among the few who engaged, there wasn’t a significant difference in engagement according to whether the letter was reframed with insights from behavioral science. This suggests that biggest opportunity is around how to engage and capture the attention of health care leaders. This is a much bigger opportunity that trying to optimize messaging once engaged.

c. Much more emphasis of the discussion should be focused on what is known re engagement/open rates of letters/emails for engaging health care leaders and clinicians. The findings should be placed in this context and opportunities to optimize in the future including the messenger, subject line, and steps to decrease friction to reaching web resources (getting rid of login steps).

d. I suspect that one factor associated with low engagement in Dec 2019 and Jan 2020 among health care leaders was that there was a deluge of email communications and meetings occurring around that time with regards to planning for the Covid-19 preparedness and also December and January are difficult times to engage health care leaders since this is the beginning of budget planning season and there many high priority meetings stacked just before and after the holidays. It may be worth repeating at other times of year outside of the emergence of a new pandemic.

e. It seems like a major missed opportunity here was a lack used-centered design and feedback from health care leaders and pilot testing of alternative nudge strategies before full deployment. Little details of design can have outsized impacts. This should be highlighted as opportunity for future optimization.

Reviewer #2: The study explores the efficacy of a behavioral nudge in influencing healthcare leaders.

A total of 2387 healthcare leaders in the United States were randomized to receive one of two types of cover letters. Both letters provided a link to a resource website and a report with survey results, but the "nudge" letter included additional elements like text color and icons to highlight how an organization’s results compared to the peers.

There were no statistically significant effects of the intervention on website access, and regardless of cover letter format, website access was low, with less than 2% of all organizational leaders targeted, clicking through to website resources.

According to the authors, the qualitative results suggest, however, that providing administrative leaders with data comparing their organization’s performance to that of peers can spur follow-up action aimed at promoting evidence-based practice adoption.

Study Design and Methodology:

The pragmatic parallel-group randomized controlled trial approach is appropriate for the study's objectives. However, the manuscript would benefit from elaborating on the selection criteria for these leaders.

The distinction between the "nudge" letter and the control letter should be more pronounced in the manuscript. A clearer explanation of the specific elements used in the nudge letter (e.g., text color, icons) and -especially - how they are expected to influence behavior, would be beneficial.

The manuscript should include a more comprehensive discussion on the implications of the relatively low response rate (1.8% for the nudge letter and 1.4% for the control letter).

Literature Review and Theoretical Framework:

The introduction provides a good foundation for the study, highlighting the challenges in integrating evidence-based practice in healthcare and the role of behavioral nudges. However, it could be strengthened by incorporating more recent literature to establish the current state of research in this field. This would position the study more firmly within the existing body of knowledge.

The manuscript should explore in greater depth the theoretical underpinnings of nudging in the context of healthcare leadership. Discussing theories of behavioral economics in relation to healthcare leadership decision-making could provide a richer theoretical framework.

Results and Discussion:

The discussion section is unsatisfactory at the moment. It needs to delve deeper into the implications of the results. Specifically, a more critical analysis of why the nudges did not significantly increase website access is needed.

The qualitative insights from the interviews are valuable but are not sufficiently integrated into the overall discussion. A more thorough analysis of these insights in relation to the quantitative findings would offer a more nuanced understanding of the study's implications.

Technical Aspects and Presentation:

The manuscript generally follows a clear structure, but there are areas where the flow can be improved for better readability. For example, transitions between sections, particularly between the methodology and results sections, could be smoother.

In summary, the manuscript presents an interesting study with potential implications for healthcare leadership and practice. However, to fully realize its contribution to the field, the above-mentioned revisions are necessary. The focus should be on strengthening the theoretical framework and deepening the analysis of the results (both quantitative and qualitative).

I would also suggest to extend the Conclusions section of the article.

6. PLOS authors have the option to publish the peer review history of their article (what does this mean?). If published, this will include your full peer review and any attached files.

Reviewer #1: No

Reviewer #2: No

---

## [Author Response · Author response to Decision Letter 0]

4 Mar 2024

I have attached a letter that responds to reviewer comments.

---

## [Decision Letter · Decision Letter 1]

23 Aug 2024

PONE-D-23-33529R1Assessing a behavioral nudge on healthcare leaders’ intentions to implement evidence-based practicesPLOS ONE

Dear Dr. Crawford,

Thank you for submitting your manuscript to PLOS ONE. After careful consideration, we feel that it has merit but does not fully meet PLOS ONE’s publication criteria as it currently stands. Therefore, we invite you to submit a revised version of the manuscript that addresses the points raised during the review process.

**ACADEMIC EDITOR:** **Please address the reviewers' feedbacks and suggestions to further improve your work.**

We look forward to receiving your revised manuscript.

Kind regards,

Laura Brunelli, MD, PhD

Academic Editor

PLOS ONE

Journal Requirements:

Reviewers' comments:

Reviewer's Responses to Questions

**Comments to the Author**

1. If the authors have adequately addressed your comments raised in a previous round of review and you feel that this manuscript is now acceptable for publication, you may indicate that here to bypass the “Comments to the Author” section, enter your conflict of interest statement in the “Confidential to Editor” section, and submit your "Accept" recommendation.

Reviewer #1: All comments have been addressed

Reviewer #3: (No Response)

2. Is the manuscript technically sound, and do the data support the conclusions?

Reviewer #1: Yes

Reviewer #3: Yes

3. Has the statistical analysis been performed appropriately and rigorously? 

Reviewer #1: Yes

Reviewer #3: Yes

4. Have the authors made all data underlying the findings in their manuscript fully available?

Reviewer #1: No

Reviewer #3: Yes

5. Is the manuscript presented in an intelligible fashion and written in standard English?

Reviewer #1: Yes

Reviewer #3: Yes

6. Review Comments to the Author

Reviewer #1: The authors did a very nice job responding to the reviewer critiques and suggestions. This will be a nice contribution to the literature.

Reviewer #3: The authors report on a randomized control trial undertaken to test the effectiveness of focused communication strategies, informed by behavioural nudge theory, in influencing healthcare leaders to adopt evidence-based care delivery practices. The novel aspect of this study is the focus on healthcare leaders as opposed to clinicians given that these leaders often have the decision-making authority to promote and oversee the implementation of improvements. The methods were described in sufficient detail and were appropriate for the study aims. Ethics approval for the study was noted. The statistical analysis was appropriate given the limited data to be analyzed. The qualitative analysis of the transcribed interviews was explained briefly but with sufficient detail given the purpose of the study. The discussion flowed logically and considered important ways to understand the findings. The limitations were realistic and demonstrated important insights into the challenges experienced in conducting the study. Despite the very low response rate for the quantitative measure to assess the intervention outcome (visiting the website resources) and the non-significant statistical results, the report offers many insights into the issue of how organizations adopt evidence-based change and likewise how to study the impact of strategies to promote adoption through the use of nudge theory. Overall, the study report is well written and was a pleasure to read.

While there are no critical gaps in the report, a few suggestions are offered for consideration only. They reflect my limited knowledge of nudge theory and the desire to understand some aspects of the study in more detail. Some minor edits are also mentioned.

Introduction – It might be useful to add more detail about nudge theory to briefly explain the mechanism by which people (i.e., leaders) are influenced. As I read the paper, I also wondered how nudge-based strategies fit (or not) into implementation science concepts.

Table 2 – I was confused by the use of the notations ‘a’ and the asterisk as the explanation at the bottom of the table does not differentiate ‘a’ from ‘*’. As it is currently stated, only an asterisk is needed.

Table 3 – The qualitative themes and exemplars are described clearly. I was left wondering if there were any comments about the nature or quality of the nudge messaging specifically. Most the themes were about actions but not the qualities of the actual intervention (social norm messaging and comparison charts).

Discussion – It would be useful to link back the original aim and mention nudge theory or strategies. The word ‘nudge’ only appears twice on page 15 but nowhere else in the discussion section onward. Since nudge theory played an important role in the design of the intervention (and it is in the report title), it would be useful to explicitly state what has been learned about nudge-based strategies and this field of study.

Figure 1 – The final box under each arm of the randomized groups (bottom of diagram) seems redundant i.e. ‘allocated to control letter’ and ‘included in the analysis’ are the same numbers. The diagram could logically end with the box for ‘received allocated intervention’.

7. PLOS authors have the option to publish the peer review history of their article (what does this mean?). If published, this will include your full peer review and any attached files.

Reviewer #1: No

Reviewer #3: No

---

## [Author Response · Author response to Decision Letter 1]

16 Sep 2024

Below are responses to reviewer comments. These are also included in the attached "Response to Reviewer Comments" document.

Reviewer Comments

Reviewer #1: The authors did a very nice job responding to the reviewer critiques and suggestions. This will be a nice contribution to the literature.

Reviewer #3: The authors report on a randomized control trial undertaken to test the effectiveness of focused communication strategies, informed by behavioural nudge theory, in influencing healthcare leaders to adopt evidence-based care delivery practices. The novel aspect of this study is the focus on healthcare leaders as opposed to clinicians given that these leaders often have the decision-making authority to promote and oversee the implementation of improvements. The methods were described in sufficient detail and were appropriate for the study aims. Ethics approval for the study was noted. The statistical analysis was appropriate given the limited data to be analyzed. The qualitative analysis of the transcribed interviews was explained briefly but with sufficient detail given the purpose of the study. The discussion flowed logically and considered important ways to understand the findings. The limitations were realistic and demonstrated important insights into the challenges experienced in conducting the study. Despite the very low response rate for the quantitative measure to assess the intervention outcome (visiting the website resources) and the non-significant statistical results, the report offers many insights into the issue of how organizations adopt evidence-based change and likewise how to study the impact of strategies to promote adoption through the use of nudge theory. Overall, the study report is well written and was a pleasure to read.

While there are no critical gaps in the report, a few suggestions are offered for consideration only. They reflect my limited knowledge of nudge theory and the desire to understand some aspects of the study in more detail. Some minor edits are also mentioned.

Introduction – It might be useful to add more detail about nudge theory to briefly explain the mechanism by which people (i.e., leaders) are influenced. As I read the paper, I also wondered how nudge-based strategies fit (or not) into implementation science concepts.

Response: We appreciate this suggestion and agree that additional context about the mechanisms by which nudges influence individuals would be helpful to include in the Introduction. We have therefore added in a few sentences to better describe this process, as well as how nudge-based strategies fit into the larger field of implementation science:

Introduction, Page 4, Lines 110 - 116: “Nudges can help those interested in implementing evidence-based practices overcome organizational constraints. They do this by altering aspects of one’s decision-making environment (known as “choice architecture”)(6) to make implementation feel easier or more salient; this can be done by changing the placement or attributes of objects or modifying messaging.(5, 29, 30) Nudging is therefore one strategy among many to promote or strengthen implementation science efforts, which encourage the uptake of evidence-based practices to improve the quality and efficacy of health services.(31, 32)” 

Table 2 – I was confused by the use of the notations ‘a’ and the asterisk as the explanation at the bottom of the table does not differentiate ‘a’ from ‘*’. As it is currently stated, only an asterisk is needed. 

Response: The “a” referenced above appears to be a formatting glitch that appeared for the reviewer. An “a” does not appear in our version of Table 2; we currently only include asterisks and no letter-based superscripts.

Table 3 – The qualitative themes and exemplars are described clearly. I was left wondering if there were any comments about the nature or quality of the nudge messaging specifically. Most the themes were about actions but not the qualities of the actual intervention (social norm messaging and comparison charts).

Response: Thank you for this suggestion. Yes, we did ask interviewees who received a nudge letter about their perceptions of the nudge messaging and data display. We have now added in additional comments that reflect these reactions:

Table 3, Page 15: 

Specific Findings: “Found the nudge messaging and peer comparison data impactful or persuasive.” (nudge letter recipients) (nudge=6/8)

Exemplar Quotes: “Seeing a "no" response when peers had a "yes" was a motivator” (Physician practice leader, nudge letter). 

“My eyes immediately went to the thumbs up and the red triangle.” (Hospital leader, nudge letter). 

Discussion – It would be useful to link back the original aim and mention nudge theory or strategies. The word ‘nudge’ only appears twice on page 15 but nowhere else in the discussion section onward. Since nudge theory played an important role in the design of the intervention (and it is in the report title), it would be useful to explicitly state what has been learned about nudge-based strategies and this field of study.

Response: We appreciate and agree with this suggestion and have worked to better link the Discussion section back to nudge theory, which we described in the Introduction: 

Discussion, Page 19, Lines 375 – 381: “The fact that our study subjects – data-driven, outcomes-oriented, business-minded individuals – appeared more influenced by the peer comparison data demonstrates the importance of accounting for audience characteristics and situational context when applying nudge theory and designing nudge strategies. In retrospect, use of peer comparison data seems highly appropriate for leaders who need to make reflective professional decisions; social norms messaging and iconography – which produce automatic, almost subconscious reactions – may be more effective nudge strategies for individuals who need to make quicker, simpler decisions.”

Figure 1 – The final box under each arm of the randomized groups (bottom of diagram) seems redundant i.e. ‘allocated to control letter’ and ‘included in the analysis’ are the same numbers. The diagram could logically end with the box for ‘received allocated intervention’. 

Response: In the initial set of comments, a reviewer suggested that we replace Figure 1 with Figure S1, which provided a more clear and straightforward visualization of the study steps. We agreed with this suggestion and made the replacement. The current reviewer is referencing the original Figure 1, which we thought we had removed in the last round of edits. We apologize if we inadvertently uploaded the wrong version of Figure 1. We will be sure to upload the correct version, which does not have any redundancies and no longer has a box labeled “included in the analysis.”

---

## [Editor Report · Decision Letter 2]

19 Sep 2024

Assessing a behavioral nudge on healthcare leaders’ intentions to implement evidence-based practices

PONE-D-23-33529R2

Dear Dr. Crawford,

We’re pleased to inform you that your manuscript has been judged scientifically suitable for publication and will be formally accepted for publication once it meets all outstanding technical requirements.

Kind regards,

Laura Brunelli, MD, PhD

Academic Editor

PLOS ONE
---

## [Editor Report · Acceptance letter]

14 Nov 2024

PONE-D-23-33529R2 

PLOS ONE

Dear Dr. Crawford, 

I'm pleased to inform you that your manuscript has been deemed suitable for publication in PLOS ONE. Congratulations! Your manuscript is now being handed over to our production team.

Kind regards, 

on behalf of

Dr. Laura Brunelli 

Academic Editor

PLOS ONE